# Spatial Differentiation of Non-Grain Production on Cultivated Land and Its Driving Factors in Coastal China

**Yan Sun [1], Yuanyuan Chang [2], Junna Liu [2], Xiaoping Ge [3], Gang-Jun Liu [4] and Fu Chen [1,2,\*]**

[1]  College of Public Policy and Management, Hohai University, Nanjing 210098, China; suny@hhu.edu.cn
[2]  School of Public Policy, China University of Mining and Technology, Xuzhou 221043, China; changyy@cumt.edu.cn (Y.C.); liujunna@cumt.edu.cn (J.L.)
[3]  College of Hydrology and Water Resources, Hohai University, Nanjing 210098, China; njgexp@hhu.edu.cn
[4]  Geospatial Science, School of Science, STEM College, RMIT University, Melbourne 3000, Australia; gang-jun.liu@rmit.edu.au
[\*]  Correspondence: chenfu@cumt.edu.cn; Tel.: +86-5168-388-3501

**Abstract:** The rapid development of urbanization and industrialization in coastal China in the past 20 years has exerted a huge squeezing effect on agricultural land use. The phenomenon of non-grain production on cultivated land (NGP) is very common, seriously threatening the protection of high-quality arable land and national food security. In order to find out the overall situation regarding NGP on cultivated land in coastal China, this study revealed the spatial differentiation of NGP and its main driving factors by spatial autocorrelation analysis, multiple linear regression models and geographically weighted regression analysis (GWR). The results show that: (1) in 2018, the non-grain cultivated land area of 11 provinces along the coast of China was about $15.82 \times 10^6$ hm$^2$, accounting for 33.65% of the total cultivated land area. (2) The NGP rate in 11 provinces gradually decreased from south to north, but the NGP area showed two peak centers in Guangxi province and Shandong province, then decreased gradually outwards. (3) The low economic benefit of the planting industry (per capita GDP and urban-to-rural disposable income ratio) was the most important driving force, leading to the spatial differentiation of NGP, while the number of rural laborers and land transfer areas also acted as the main driving factors for the spatial differentiation of NGP. However, the influence of each driving factor has obvious spatial heterogeneity. The non-grained area and the non-grain production rate at the municipal level were completely different from those at the provincial level, and the spatial heterogeneity was more prominent. In the future, the local government should control the disorganized spread of NGP, scientifically set the bottom line of NGP, reduce the external pressure of NGP, regulate multi-party land transfer behavior, and strengthen land-use responsibilities. This study can provide a scientific foundation for adjusting land-use planning and cultivated land protection policies in China and other developing countries.

**Keywords:** non-grain production; cultivated land protection; spatial autocorrelation; geographically weighted regression (GWR); driving factor

## 1. Introduction

China has been in a stage of rapid industrialization and urbanization since 1990, resulting in the non-agriculturalization of $2.94 \times 10^5$ hm$^2$ of cultivated land every year [1,2], which directly threatens food security in China. In fact, in addition to the dominant loss of cultivated land, some high-yield cash crops are driven by economic interests, encouraging farmers to replace grain crops on a large scale [3]. The recessive loss of grain production capacity, also referred to as NGP, should not be ignored. In 2019 alone, China's grain-planted acreage dropped by $9.70 \times 10^5$ hm$^2$, which not only seriously threatened national food security but also caused huge risks to rural social stability, the agricultural landscape and ecosystem health [4–7]. How to properly solve the problem of

NGP has become the focus of attention of the government, academia and the public [8–11]. In addition, the influx of industrial and commercial capital into rural areas in recent years has also accelerated NGP [12,13]. Cultivated land protection is the cornerstone of food security, and emergencies such as the global spread of COVID-19, the acceleration of climate change, and the blockading of international food trade have further demonstrated that restraining NGP is urgent.

NGP has been a common phenomenon of land-use conversion on a global scale. NGP is different from non-agriculturalization, does not cause the loss of cultivated land [14], and is different from farmland marginalization [15]. Generally, NGP refers to the phenomenon whereby the farmland intended for grain cultivation is occupied by cash crops, forests, fruits or the stock-breeding industry. NGP is mainly attributable to the low economic benefit of planting grain, and is also affected by regional resource endowments, economic development, social factors and agricultural production habits. European and American countries rarely intervene in land-use conversion within the agricultural sector, and mainly rely on the market to adjust the planting structure. The United States, based on its international and domestic food supply and demand conditions, formulated a lower limit for the planting scale of major crops every year, which not only guaranteed food security but also fully protected the farmland ecology and high-efficiency utilization of farmland resources [16]. Most EU countries and Japan adjusted their fallow scale according to market supply and demand conditions, to cope with fluctuations in international food prices [17–20]. The agricultural planting structure in some developing countries such as Brazil is not only affected by their own needs but also by the investment motives of multinational companies [21]. The Chinese government discourages NGP because it needs to use 8% of the world's cultivated land to feed 18.4% of the world's population.

Food security is the primary consideration of the Chinese government. The Chinese government has long discouraged NGP and issued an order prohibiting NGP on 17 November 2020 (http://www.gov.cn/zhengce/content/2020-11/17/content_5562053.htm). The General Office of the State Council issued the "Opinions on Preventing Non-Grain Production on Cultivated Land and Stabilizing Grain Production", clearly understanding the profound urgency of preventing NGP and stabilizing grain production, earnestly grasping the national food security initiative, and establishing the reporting system of NGP so as to keep it strictly controlled. However, it is very difficult to enforce the policy ban regarding NGP. Firstly, both local governments and farmers lack motivation because of the low economic benefits of crop plantation. Some researchers have validated the drive function of the low economic benefits of crop plantation on NGP and analyzed the negative effect of NGP on national food security [22–25]. Secondly, the land transfer and profit-seeking of industrial and commercial capital has promoted the conversion to NGP. Seto et al. found that the pulling of foreign investment, unscientific planning, and improper management have led to a serious loss of cultivated land and grain production capability in the Pearl River Delta [26]. Liao et al. analyzed the effects of the profit-seeking behaviors of contractors on NGP in the land transfer process and proposed related measures and suggestions for preventing excess NGP, from the perspective of restricting improper land transfer [27]. Thirdly, unlike non-agriculturalization, NGP did not directly lead to the loss of cultivated lands, with less hazard. NGP can be mostly regarded as an internal structural adjustment of agriculture and cannot be completely prohibited because of the involvement of farmers' interests. In addition, certain specific factors, including cultivation scale [28–30], industrial structure [31], family characteristics [32–34] and benefit farming policies [35] have also affected NGP in China.

Remote sensing and UAV technology have provided an effective means of monitoring NGP [36]. Some studies have focused on the spatial characteristics of cultivated land, non-agriculturalization, and farmland marginalization [37,38]. However, its workload and cost are huge. Remote-sensing data with a resolution greater than 1 m is also not suitable for monitoring specific crops such as garlic and wheat, which cannot be

distinguished at all in the winter. Currently, it is almost impossible to use remote-sensing technology to accurately monitor NGP on a national scale with a country as large as China. Authoritative statistical data is indispensable for studying the spatial characteristics of NGP on a national scale. Moreover, cultivated land is widely distributed, with large differences in natural, social and economic conditions across China. The behavior and causes of NGP in different regions are not the same. On such a large scale, clarifying the spatial pattern of NGP and its causes, in terms of the current, specific, cultivated land protection and national food security is particularly important.

Although previous studies have encompassed many issues regarding NGP, NGP is a result of complex behavior involving multiple factors and myriad stakeholders [39–42]. The spatial scale is particularly important for the management and interpretation of NGP. At present, the literature focusing on the spatial pattern differentiation of NGP and its causes at different scales is almost blank, which is unfavorable for the governance of NGP behavior and human-land-grain production nexus under the conditions of economic and social development differences. Therefore, 11 provinces and 130 cities in coastal China were selected as the research objects in this study, because they are most severely affected by the low economic efficiency of grain planting, the intervention of industrial and commercial capital, and urbanization. The aims of this study are following as: (1) Explore the spatial pattern and differentiation characteristics of non-grain farmland scales at the provincial scales and municipal scales; (2) Identify the influencing factors of non-grain farmland; (3) Interpret the driving mechanism of non-grain farmland. This study is expected to provide a scientific foundation for global food security and farmland protection decision-making.

## 2. Materials and Methods

### 2.1. Concept Definition and Research Framework

2.1.1. Concept Definitions

The definition and calculation methods of NGP have always been very vague in China. Some studies have used indicators such as "grain-to-grain ratio" [43], "non-grain sown area to crop sown area" [44,45], "non-food cultivation and circulating farmland area to circulating farmland area" [46] to describe the rate of NGP, but these concepts are not very accurate. According to the documents issued by the Ministry of Natural Resources (http://www.gov.cn/zhengce/zhengceku/2021-04/12/content_5599101.htm (accessed on 1 April 2021)) the gain crops in this study only include three types of rice, wheat, and corn. All the farming behaviors of planting other crops on cultivated land except for three types of rice, wheat, and corn are defined as NGP. The area and ratio of NGP can be defined as follows:

$$A_{ngc}^i = A_t^i \times C^i - A_c^i \tag{1}$$

$$R_{ngc}^i = 1 - A_c^i / \left( A_t^i \times C^i \right) \tag{2}$$

where $A_{ngc}^i$ denotes the NGP area of the *i*-th unit, $A_t^i$ denotes the cultivated area of the *i*-th unit, $C^i$ denotes the multiple crop index of the *i*-th unit, $A_c^i$ denotes the grain-sowing area of the *i*-th unit, and $R_{ngc}^i$ denotes the NGP rate of the *i*-th unit. The cultivated area, the grain-sowing area, and the multiple crop index of each municipal unit are directly derived from the *2019 Statistical Yearbooks* for each municipal unit. The advantage of the definition of NGP used in this study is that the aforementioned three indicators are furnished with accurate and authoritative statistical data in China; at the same time, it also encounters the confounding factors of the existence of arable land and facility agricultural land in vegetable production, and the existence of arable land and garden land in fruit production. Therefore, we can only rely on authoritative statistical data to monitor NGP in China. In addition, considering the multiple cropping index also brings the rate of NGP closer to the actual situation.

2.1.2. Research Framework

Multiple variables, including economy, society, and resource endowment, have inexhaustible links with NGP, and the interaction of different elements together drives the situation of NGP in China. The research framework includes the following steps (Figure 1). Step 1: the related datasets of NGP were established so as to get a clear knowledge of both the quantity and spatial distribution of NGP. Step 2: the spatial pattern and distribution characteristics of NGP, and the spatial distribution, the global agglomeration degree, and the local agglomeration difference were analyzed with ArcGIS 10.2 (ESRI, Sacramento, CA, USA) and Stata 17.0 (StataCorp, Houston, TX, USA). Step 3: the driving mechanism and regional difference of NGP were identified and the main influencing factors were explored via the use of a multiple linear regression model. Moreover, the variation of the main driving factors of NGP according to the local geographic position was analyzed using the GWR model. Finally, related policy suggestions on controlling or reducing NGP were developed.

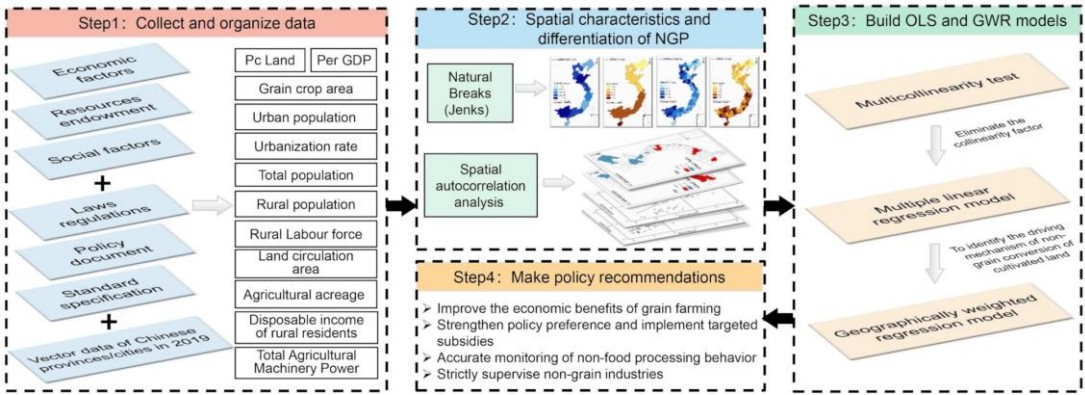

**Figure 1.** Research framework of this study.

*2.2. The Study Area, Variable Selection and Data Processing*

2.2.1. The Study Area

Coastal China has 11 provincial units, including Liaoning, Hebei, Tianjin, Shandong, Jiangsu, Shanghai, Zhejiang, Fujian, Guangdong, Guangxi, and Hainan, with a total area of 950,000 km² (Figure 2). The coastal area straddles the tropical, subtropical and temperate zones, and belongs to a typical maritime monsoon climate. The annual precipitation is 750–1700 mm, decreasing from south to north. In 2020, the coastal area had a population of approximately 643 million, accounting for 44.5% of the total population of China, with an urbanization rate of over 60.0% in 130 municipal units, and a per capita GDP close to USD 20,000, including almost all of China's most economically developed provincial units. The cultivated land per capita is small, at only 0.08 ha in coastal China. Being adversely affected by the low economic efficiency of the farming industry, the intervention of industrial and commercial capital, and urbanization, the situation regarding grain planting in coastal China is not good, and the phenomenon of NGP has become very common.

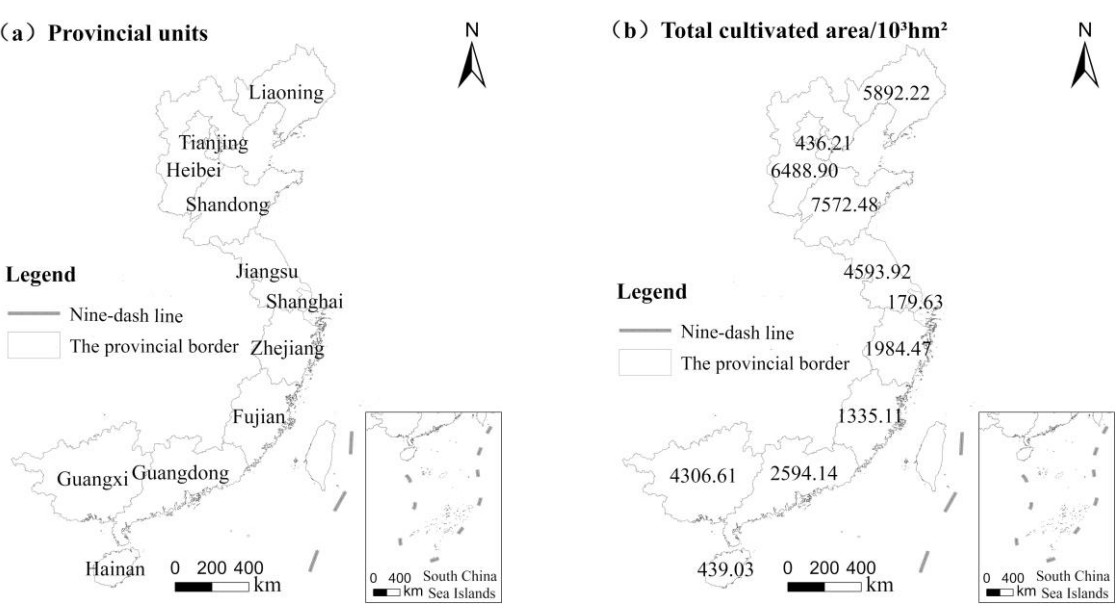

**Figure 2.** The study area: (**a**) location map and (**b**) cultivated land area in each province.

2.2.2. Selection of Variables

This study refers to previous research results, based on the research framework and combined with the availability of data and taking into account classic variables, to determine the primary driving factors for NGP. These factors can better explain the impetus driving NGP from different viewpoints, such as economic level, social development, resource allocation, and policy regulation (Table 1). Label $x_1$ represents the impact of economic development on NGP [47,48], $x_2$ represents the impact of industrial and commercial capital on NGP [49–52], $x_3$ represents the impact of low planting efficiency on NGP [53,54], $x_4$, $x_5$ and $x_6$ represent the impact of resource endowments on NGP [55], and $x_7$ represents the impact of material capital on NGP [56]. Labels $x_8$, $x_9$, $x_{10}$, $x_{11}$, and $x_{12}$ represent the impact of human capital on NGP [57], while $x_{13}$ represents the impact of land transfer on NGP [58], and $x_{14}$ represents the impact of internal organizations or local policies on NGP [59–61].

**Table 1.** Preliminarily selected driving factors of NGP in this study.

| Indexes | Factors | Unit | Label | Indexes | Factors | Unit | Label |
|---|---|---|---|---|---|---|---|
| Economic indexes | Per capita GDP | CNY | $x_1$ | Social indexes | Agricultural machinery power | $10^4$ kW | $x_8$ |
| | Actual use foreign investment | $10^6$ \$ | $x_2$ | | Urban population | per | $x_9$ |
| | Urban-to-rural disposable income ratio | / | $x_3$ | | Rural population | per | $x_{10}$ |
| Resource endowment | > 10 °C accumulated temperature | °C | $x_4$ | | Rural labor force | per | $x_{11}$ |
| | Slope | % | $x_5$ | | Urbanization rate | % | $x_{12}$ |
| | Grain-sowing area | hm² | $x_6$ | Policy indexes | Land transfer area | hm² | $x_{13}$ |
| | Cultivated area per rural labor | hm²/per | $x_7$ | | Agricultural cooperatives number | Number | $x_{14}$ |

2.2.3. Data Source

Population, GDP, rural labor, residents' disposable income, urbanization rate, and the cultivated land area of 11 provinces and 130 cities in coastal China were collected in

this study. Then, indicators such as the per capita GDP and the disposable income ratio of urban and rural residents were calculated. The data were sourced from the *Chinese Statistics Yearbook of 2019* (http://www.stats.gov.cn/tjsj/ndsj/ (accessed on is 24 September 2019)), *Chinese Land & Resource Yearbook of 2019* (https://www.yearbookchina.com/navibooklist-n3020013365-1.html (accessed on 1 October 2019.)), and various *Statistics Yearbooks* (https://www.yearbookchina.com/navibooklist-n3020013060-1.html (accessed on 12 October 2019)) of different provinces and cities. Moreover, certain indexes that included the per capita GDP and urban-to-rural disposable income ratio were also calculated. Other data including the sowing area of crops, the sowing area of grains, the number of agricultural cooperatives, and the total power of agricultural machinery were considered. Those data were sourced from the *Data Bulletins of the 3rd National Agricultural Census* (http://www.stats.gov.cn/tjsj/tjgb/nypcgb/ (accessed on 16 December 16 2017.)) of 130 cities and *The Statistical Bulletin of National Economic and Social Development*, based on which the cultivated area per capita of rural labor was calculated. The actual utilized foreign investment of agriculture was sourced from the *Chinese Foreign Economic and Trade Yearbook in 2019* (https://www.yearbookchina.com/navibooklist-n3020031901-1.html (accessed on 25 October 2019)), and the land transfer area was sourced from the rural property rights trading information service platform by the Ministry of Agriculture and Rural Affairs of China.

### 2.3. Research Method

2.3.1. Spatial Auto-Correlation Analysis Method

Global Moran's I, denoted as $I_g$, can directly reflect the existence of a correlation between a region and the neighboring regions, in terms of the spatial distribution of a property and the related correlation degree [47]. $I_g$ falls into the range of [–1, 1]. A positive value of $I_g$ suggests aggregated distribution, accompanied by a positive correlation between spatial factors, and a negative value of $I_g$ is indicative of discrete distribution, with a negative correlation between spatial factors. When $I_g$ = 0, the spatial factor follows a random distribution without correlation. However, Global Moran's I can only reflect the overall distribution of spatial factors and cannot measure local agglomeration and spatial heterogeneity. Accordingly, the Local Moran's I, denoted as $I_l$, is introduced for analyzing the difference between the spatial factor and the neighboring regions [47,62]. When $I_l > 0$, the observed value of the *i*-th unit shows a significantly slight difference from the surrounding units, i.e., a high observed value is surrounded by high values, or a low observed value is surrounded by low values. When $I_l < 0$, the observed value of the *i*-th unit shows a significant difference from the surrounding units, i.e., a low observed value is surrounded by high values or a high observed value is surrounded by low values. To be specific, $I_g$ and $I_l$ can be calculated as:

$$I_g = \frac{n \cdot \sum_{i=1}^{n} \sum_{j=1}^{n} W_{ij}(x_i - x)(x_j - x)}{\sum_{i=1}^{n} \sum_{j=1}^{n} W_{ij} \sum_{i=1}^{n}(x_i - \overline{x})^2} \tag{3}$$

$$I_l = \frac{(x_i - \overline{x}) \sum_{j=1}^{n} W_{ij}(x_j - \overline{x})}{\frac{1}{n} \times \sum_{i=1}^{n}(x_i - \overline{x})^2} \tag{4}$$

where $n$ denotes the number of research units; $x_i$ and $x_j$ are the observed values of the *i*-th and the *j*-th units; $\bar{x}$ denotes the mean observed value of all units; and $W_i$ denotes the spatial weight matrix of each unit. If unit *i* is adjacent to unit *j*, the weight equals 1; otherwise, the weight equals 0.

2.3.2. Ordinary Least Square (OLS) Linear Regression Analysis

Firstly, the multi-nonlinearity was diagnosed. A multi-collinearity test was performed on the independent variables by a variance inflation factor (VIF). It is generally

acknowledged that VIF = 10 is a standard value. When VLF > 10, the independent variables in the regression model show serious collinearity. The independent variables with VLF > 10 were removed for further stepwise regression and optimal solution, until VLF < 10 for all independent variables. Next, the following multiple linear regression was performed:

$$y = \beta_0 + \beta_1 x_1 + \beta_2 x_2 + \cdots + \beta_k x_k + \varepsilon \tag{5}$$

where $y$ denotes the dependent variables; $x_1 \cdots$ and $x_k$ are independent variables; $\beta_0$ is a constant; $\beta_1 \cdots$ and $\beta_k$ are the regression coefficients; and $\varepsilon$ denotes the random error. Finally, the independent variables are ranked in accordance with their importance. The contribution degrees of different independent variables on the coefficient of determination, $R^2$, can reflect the degree of contribution to the variance of dependent variables. Generally, a larger value of $R^2$ is indicative of greater effect on and more importance to the dependent variable.

### 2.3.3. Geographically Weighted Regression (GWR)

GWR is the development of the ordinary least squares (OLS) test. The OLS regression model is a global regression model. Using OLS, the estimated values of the regression coefficient is the mean value of the whole research area. Accordingly, local characteristics of the relation among variables may easily be neglected and the regression results cannot reflect the real spatial characteristics of the regression parameters. By embedding the spatial positions of the data into the regression model, GWR performs local regression based on the subsample data obtained via adjacent observation, so as to reflect the variation of the relationship among variables with the local geographic position [49,63]. This study adopted GWR to explore the spatial heterogeneity of the influencing factors of NGP in different regions of coastal China, and the related rules. The detailed formula can be written as [43,46]:

$$y_i = \beta_0(\mu_i, \vartheta_i) + \sum \beta_k (\mu_i, \vartheta_i) x_{ik} + \varepsilon_i \tag{6}$$

where $y_i$ denotes the dependent variable of the $i$-th spatial position (NGP area); $x_i$ denotes the explanatory variable of the $i$-th spatial position (the driving factor); $k$ denotes the set of the explanatory variables; $(\mu_i, \vartheta_i)$ denotes the geographic center coordinate of the $i$-th spatial position; $\beta_k(\mu_i, \vartheta_i)$ denotes the regression coefficient of the spatial unit $i$; $\varepsilon_i$ is the error item; $\beta_0$ denotes the regression constant at the spatial unit $i$. GWR generally adopts the weighted least squares test to perform local regression on each spatial unit, establishes the weighting function with a Gaussian function, calculates the bandwidth using an Akaike information criterion ($AIC_c$) and kernel density method, and finally, performs the GWR calculation. The Gaussian function can be written as:

$$\omega_{ij} = exp \left[-\left(d_{ij}/b\right)^2\right] \tag{7}$$

where $b$ denotes the bandwidth and denotes the distance between the sampling points $i$ and $j$. When observing the data at the sampling point $i$, the weights of the other sampling points can drop with the distance, $d_{ij}$, in accordance with the Gaussian curve. For the given bandwidth, $b$, the weight of the sampling point, $j$, is lower at a larger distance of $d_{ij}$; the weight of the sampling point, when at a quite great distance from point $I$, tends to be 0.

The parameters should be set before the operation of GWR model. The longitudes and latitudes of the centroids of the municipal units were set as the coordinates. The fixed Gaussian function was set as the kernel function, and the optimal bandwidth was determined via CV cross-validation [55]. The bandwidth was determined, based on the criterion of $AIC_c$. Moreover, when using the adaptive bandwidth selection method, the minimum bandwidth should be the most adaptive bandwidth in this study:

$$AICc = 2nln(\sigma) + nln(2\pi) + n\frac{n + tr(s)}{n - 2 - tr(s)} \tag{8}$$

where $n$ denotes the number of the sampling points, $\sigma$ denotes the standard error of the error estimated item, and $tr(s)$ is the function of bandwidth and the trace of the matrix, $s$. The bandwidth corresponding to a minimum value of $AIC_c$ is the optimal value.

## 3. Results

### 3.1. Overall Pattern of NGP in Coastal China

The area of NGP in 2018 reached up to $15.82 \times 10^6$ hm², which took up 33.65% of the total cultivation area. As shown in Figure 3a, the area of NGP differed obviously among different provinces. In terms of the area of NGP, Guangdong, Guangxi, Shandong, Hebei, and Jiangsu can be classified as high- and medium-value units (with an area of NGP over $1200 \times 10^3$ hm²), among which Shandong ranked the first (with an area of NGP of $2693 \times 10^3$ hm²); Shanghai, Zhejiang, Fujian, Tianjin and Liaoning can be classified as low-value units (with an area of NGP of over $800 \times 10^3$ hm²), among which Tianjin showed the lowest area of $79.1 \times 10^3$ hm². The rate of NGP dropped gradually from south to north. To be specific, Hainan exhibited the highest rate of 59.87%, while Shanghai, Zhejiang, Guangxi, Guangdong, and Fujian all exhibited a rate of over 45% (see Figure 3b). The high rates in Hainan, Guangxi, and Guangdong can mainly be attributed to favorable climatic conditions, which are suitable for the cultivation of both tropical- and subtropical fruits and commercial crops. Shanghai demonstrated an extremely low cultivation area, in which vegetables, agricultural and sideline products, and agricultural sightseeing were well developed, thereby showing a high NGP rate. Zhejiang and Fujian had small NGP areas but a high NGP rate. Zhejiang and Fujian featured low rates of per capita cultivated land, a high proportion of hilly areas and relatively developed regional economies. Therefore, farmers were likely motivated by profit and were more likely to develop characteristic agriculture and commercial crops with higher economic benefits, such as vegetables or fish culture in paddy field [59]. Overall, the NGP area was not synchronous with the NGP rate in terms of spatial distribution at the provincial level. The NGP rate dropped gradually from south to north, while the non-grain cultivated area showed two peaks at Shandong and Guangxi, weakening toward the surrounding area.

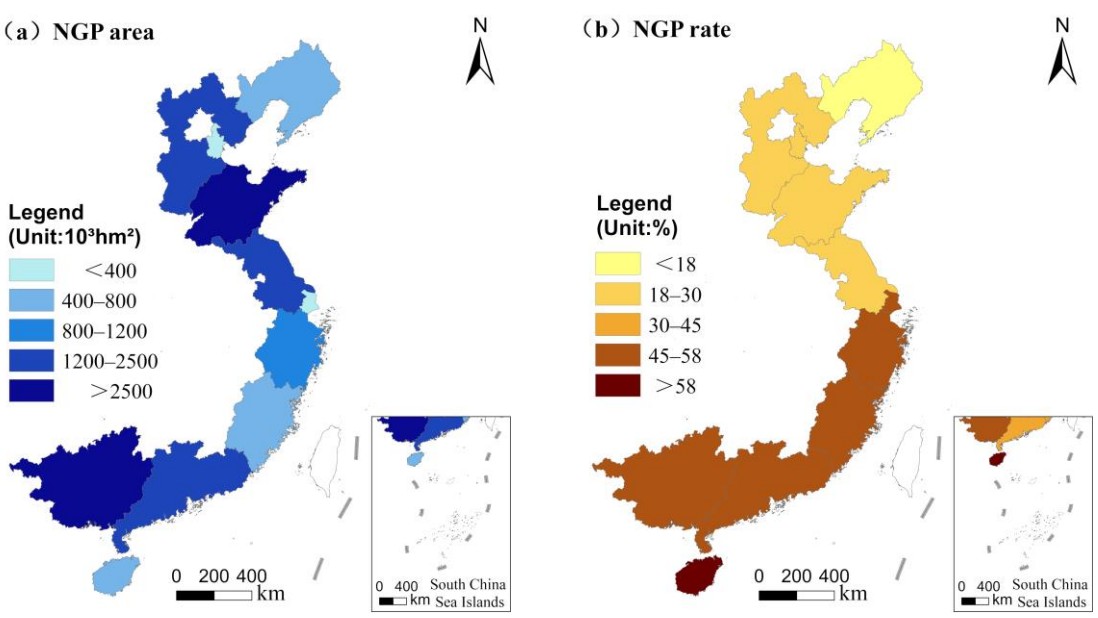

**Figure 3.** Spatial distribution of NGP in 11 provinces along coastal China in 2018: (**a**) NGP area and (**b**) NGP rate.

Both the area of NGP and the rate of NGP on a municipal scale showed different spatial patterns from the value at a provincial scale. Firstly, the area of NGP showed a dispersed distribution pattern with multiple centers. In terms of their NGP area, 11 cities can be classified as high-value cities, and are dispersed in Guangxi, Shandong and Liaoning. In total, 39 cities can be classified as low-value cities, which are relatively concentrated in eastern Liaoning, the coastal areas of Zhejiang and Fujian, the Pearl River Delta, and Hainan (Figure 4a). Secondly, the NGP rate also dropped from south to north, with greater spatial heterogeneity at a municipal level. The NGP rates of 31 municipal cities all exceeded 58%. These 31 municipal units, with NGP rates of over 58%, exhibited concentrated distribution in the vicinity of the East China Sea (Figure 4b). These units were mostly located in hilly areas, with small cultivation plots and per capita cultivation areas that were unfavorable for mechanized operation. In addition, favorable climatic conditions were suitable for the plantation of tropical and sub-tropical fruits. Overall, the NGP area and rate showed no synchronous spatial distributions at the municipal level. For example, Hainan, Zhejiang, and Fujian had high NGP rates but were low in NGP area, while Hebei, Shandong, and Jiangsu had large areas of NGP but a small actual rate of NGP. Only a few municipal units in Guangdong and Guangxi exhibited both an area of NGP and a rate of NGP.

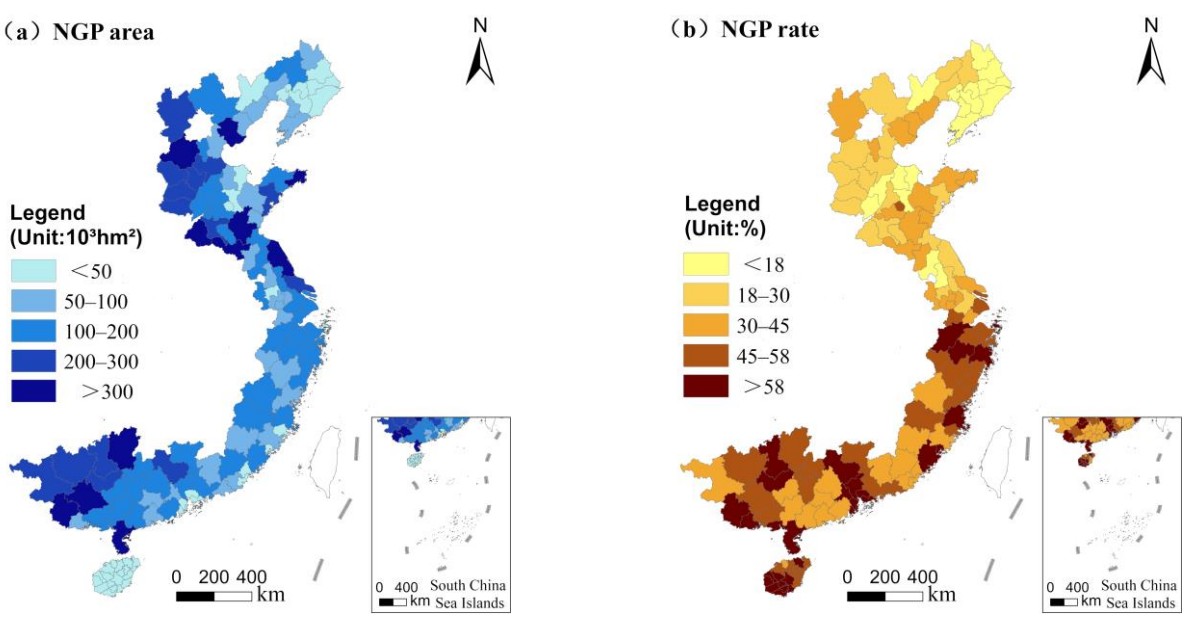

**Figure 4.** Spatial distribution of NGP in 130 cities along coastal China in 2018: (**a**) NGP area and (**b**) NGP rate.

### 3.2. Spatial Distribution of NGP in Coastal China and Its Differentiation Characteristics

Using the NGP areas and rates of 130 cities as the observational variables, the overall spatial characteristics of the NGP of 130 cities in coastal China were judged by the Global Moran's I. The Global Moran indexes ($I_g$) of the NGP area and NGP rate of 130 cities were 0.229 to 0.394, respectively, which exceeded 0 and passed the significance level test at $p < 0.001$. The unit distribution differed greatly in different quadrants. As shown in Figure 5, most of the cities were concentrated in the first and second quadrants, suggesting obvious aggregated distribution characteristics regarding the NGP area and NGP rate.

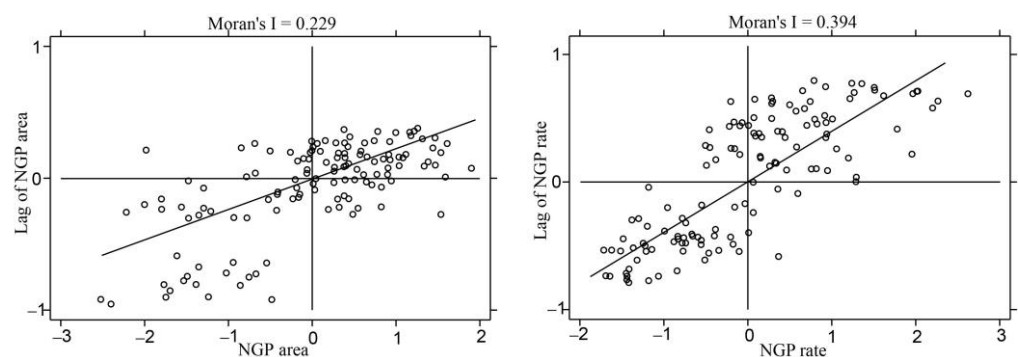

**Figure 5.** Scatter diagram of the Global Moran's I values of the NGP area and NGP rate of 130 cities in coastal China.

However, the agglomeration and mode of NGP in a specific area are still unclear, and the Local Moran's I assessment is introduced for in-depth analysis. Figure 6a shows the agglomeration of NGP areas in 130 cities of coastal China. Firstly, 2 high-high agglomeration zones, including 11 cities, can be observed. The NGP areas of these units differed slightly from the values of the adjacent units. The northern high-high

agglomeration zone was located in the Taihang Mountain area with poor natural conditions, small and dispersed fields and an unsound irrigation system. In this zone, farmers changed over to planting fruits and economic forests so as to adapt to climatic drought conditions. The southern high-high agglomeration zone featured unique natural conditions that are suitable for the large-scale development of subtropical commercial crops. Secondly, two low-low agglomeration zones, including 8 cities, can be observed. The northern low-low agglomeration zone was located on the Liaohe Plain, a traditional grain-producing region with favorable natural and cultivation conditions. In this region, the cultivated area was abundant and had few people, suggesting high suitability for large-scale mechanized grain cultivation. The southern low-low agglomeration zone was located in the Pearl River Delta, where the rice-cultivated land area was extremely low. Figure 6b shows the agglomeration of the NGP rate in 130 cities of coastal China. Three high-high agglomeration zones, including 14 cities, can be observed. The NGP rates of these cities were high and differed slightly from the values of the adjacent cities. These cities, with a superior climate, were suitable for the cultivation of subtropical fruits and commercial crops or offered hilly areas fit for the development of characteristic agriculture, thereby leading to a decline in the proportion of grain cultivation. Secondly, two low-low agglomeration zones, including 23 cities, can be observed; these were mainly located in the Liaohe Plain and the Yellow River Delta. These units, with flat terrain and a favorable climate, were suitable for the large-scale mechanized cultivation of wheat and corn. By comparing the spatial characteristics of NGP areas and rates, obvious spatial heterogeneity can be observed in high-high and low-low agglomeration zones. The cities with high NGP rates were mostly located in hilly areas with less cultivated land and a large number of commercial crops. The cities with a small NGP area were mostly distributed in the plains area and the total gross area of the arable land, accompanied by a low NGP rate.

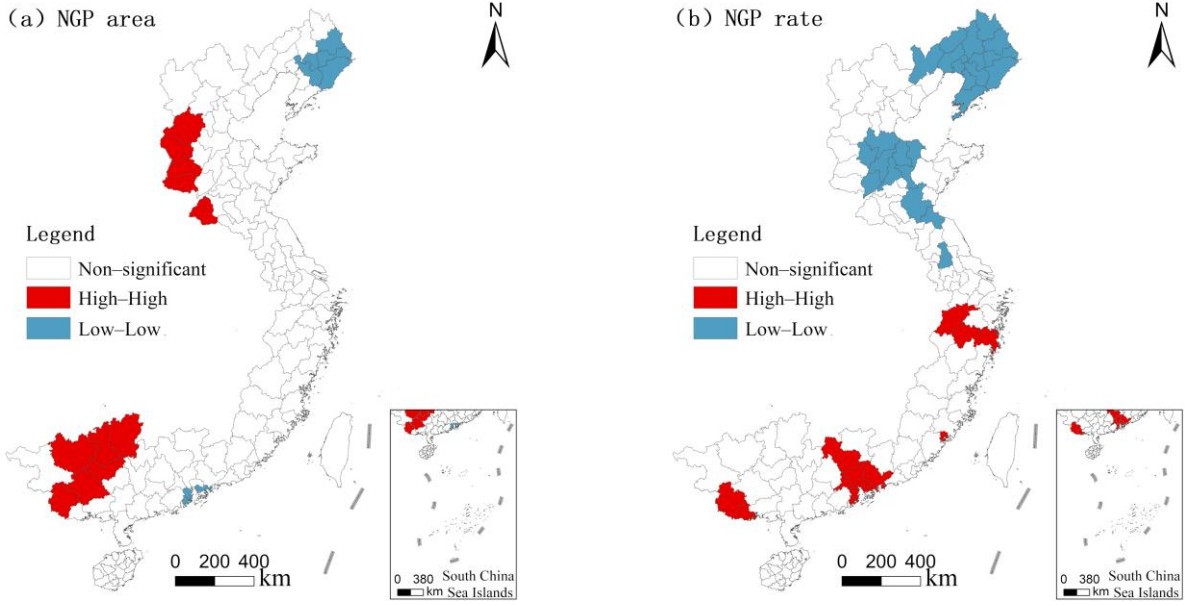

**Figure 6.** Spatial agglomeration patterns of the NGP rate and area of 130 cities in coastal China. (**a**) NGP area and (**b**) NGP rate.

*3.3. Main Driving Factors of NGP in Coastal China and Regional Differentiation*

3.3.1. Main Driving Factors of NGP in Coastal China

By considering a large number of factors that were selected preliminarily, the multicollinearity of various factors was tested. First, the variance inflation factors (VIFs) of various driving factors were calculated with SPSS18.0 (IBM, USA). The significant collinearity factors with a VIF > 10.0 were eliminated and both invalid and dominant factors were screened out. Next, a multi-factor linear equation was established via stepwise regression. It was found that the values of $R^2$ in the direct fitting models between the NGP area/rate and various driving factors were all low, suggesting that there was no great statistical significance. Then, logarithmic transformation of the independent variables was performed. However, when performing fitting on the regression model of the NGP rate, the Jarque-Bera test results were significant, suggesting that the residual cannot be described by a normal distribution, i.e., the fitting was distorted [64]. Therefore, only the logarithmic model of the NGP area was considered.

The value of $R^2$ of the regression model was 0.7149, suggesting favorable fitting results. The VIF values of various driving factors were all below 5.0, suggesting that there were no redundant variables in the model and no multi-collinearity among various factors. Meanwhile, various factors were at a high significance level, and the value passed through the significance test at a level of 0.001. This indicates that the probabilities of the effects of all dependent variables on independent variables equaled 0, i.e., the model showed statistical significance. Finally, using Stata software, these driving factors were ranked in terms of importance. Table 2 lists the effects of various driving factors on the NGP area and the related importance degrees. In terms of the degree of relative importance: urban-to-rural disposable income ratio > per capita GDP > rural labor force > land transfer area > total agricultural machinery power > rural available foreign capital investment > urbanization rate. All the above factors except the urbanization rate were positively correlated with the NGP area. The urban-to-rural disposable income ratio, per capita GDP and rural labor force were in positive correlation with the independent variables, at a significance level of 1.0%.

**Table 2.** OLS-based model fitting results between the NGP area in coastal China and its driving factors.

| Explanatory Variable | Coefficient | Standard Error | $t$ | Sig. F | Collinearity Statistics | | Importance-Based Rank | |
|---|---|---|---|---|---|---|---|---|
| | | | | | Tolerance | VIF | Contribution Degree | Ranking |
| Interception | 6.786 | 2.114 | 3.210 | 0.002 ** | | | | |
| $x_1$ | 0.239 | 0.089 | 3.293 | 0.001 *** | 0.326 | 1.767 | 0.182 | 2 |
| $x_2$ | 0.248 | 0.492 | 1.569 | 0.077 * | 0.189 | 5.295 | 0.069 | 6 |
| $x_3$ | 0.373 | 0.092 | 4.643 | 0.000 *** | 0.221 | 2.532 | 0.301 | 1 |
| $x_8$ | 0.284 | 0.176 | 1.781 | 0.072 * | 0.171 | 3.29 | 0.076 | 5 |
| $x_{11}$ | 0.129 | 0.175 | 3.215 | 0.001 *** | 0.302 | 3.267 | 0.172 | 3 |
| $x_{12}$ | −1.134 | 0.538 | −2.107 | 0.035 ** | 0.414 | 2.255 | 0.028 | 7 |
| $x_{13}$ | 0.356 | 0.368 | 1.987 | 0.059 * | 0.218 | 2.412 | 0.109 | 4 |
| OLS diagnosis | Joint F-value | | | | 0.0000 *** | | | |
| | Joint Chi-square value | | | | 212.4581, 0.0000 *** | | | |

Note: ***, ** and * show the significance at a level of 1%, 5% and 10%, respectively; the same applies to Table 3, below.

### 3.3.2. Spatial Heterogeneity of the Driving Factors of NGP in Coastal China

The average value of $I_g$ of the NGP area of 130 cities was 0.229, which indicates an obvious spatial agglomeration of NGP areas in coastal China. Therefore, the effects of various driving factors on the dependent variable space were analyzed via GWR. This study has adopted the opinion proposed by Brunsdon et al. [65]: if the value of the $AIC_c$ in the GWR-based fitting results is lower than the value in the OLS-based regression results, GWR is more effective than OLR regression in regression fitting performance and applicability. It can be observed from Table 3 that the GWR-based regression model shows a more ideal fitting performance.

**Table 3.** Comparison between OLS-based and GWR-based models regarding fitting performance.

| Model | AIC$_c$ | Goodness of Fit (R²) | Adjusted Goodness of Fit (R²) | *p* Value |
|---|---|---|---|---|
| OLS | 222.326 | 0.714 | 0.704 | 0 |
| GWR | 138.266 | 0.905 | 0.872 | 0 |

The factors with a significant correlation in the regression of 130 cities, and the related effects on the spatial heterogeneity of NGP, were analyzed and then visualized with the natural breaking-point method (Figure 7). As shown in Figure 7a, the urban-to-rural disposable income ratio exhibited a globally positive correlation with NGP areas. A larger income ratio is indicative of an area more attractive for crop production, thereby more easily promoting an increase in NGP [66]. The urban-to-rural disposable income ratio increased steadily from the northeast to the southeast, reflecting the weaker promoting efficiency of the northeast. The northeastern region was distinguished by fewer people but abundant land, a low economic level, and a small urban-rural income gap. In this region, large-scale mechanization grain cultivation instead contributed to the total economic income. As shown in Figure 7b, the per capita GDP, overall, was positively correlated with the NGP area except for a negative correlation in some municipal units of Liaoning. This suggests that the enhancement of per capita GDP promoted NGP and economic development led to active demand, diversifying agricultural products, thereby squeezing out the original grain production space [67,68]. The per capita GDP showed high absolute values in the southern and the middle regions and low values in the southeastern and northern regions, which reflects the strong correlation between per capita GDP with NGP areas in the middle municipal units. As shown in Figure 7c, the rural labor force was, overall, positively correlated with the NGP area, except in areas with a negative correlation in some municipal units of Fujian. However, this does not mean that there is a larger NGP area in municipal units with a smaller rural labor force. For most of the municipal units in the southeastern coastal region of China, except in Fujian, both cultivated areas and the rural labor force were insufficient. A greater amount of residual rural labor force imposed a greater effect on NGP, which not only shifted towards non-agricultural employment but also promoted the transfer to vegetables, forests, and fruits, with high benefits, thereby promoting NGP in terms of the agricultural structure [69–71]. Figure 7d shows the correlation between the land transfer area and the NGP area on arable land, from which an overall positive correlation can be observed. Land transfer increased the operating costs, which cannot be supported by crop cultivation. Accordingly, the transfer-ins were forced to choose NGP with a higher profit [72]. The land transfer area, overall, was higher in the southeastern coastal region and lower in the northern region, in terms of spatial distribution. This suggests that the change in NGP area induced by the change in the land transfer area in the southeastern coastal region far exceeded that in the northern region. This can be mainly attributed to hilly areas in the southeastern coastal region of China, with their small and dispersed fields. Since the hilly areas were unfavorable for the large-scale mechanized cultivation of grains, tropical- and subtropical-characteristic agricultural products were cultivated after the land transfer.

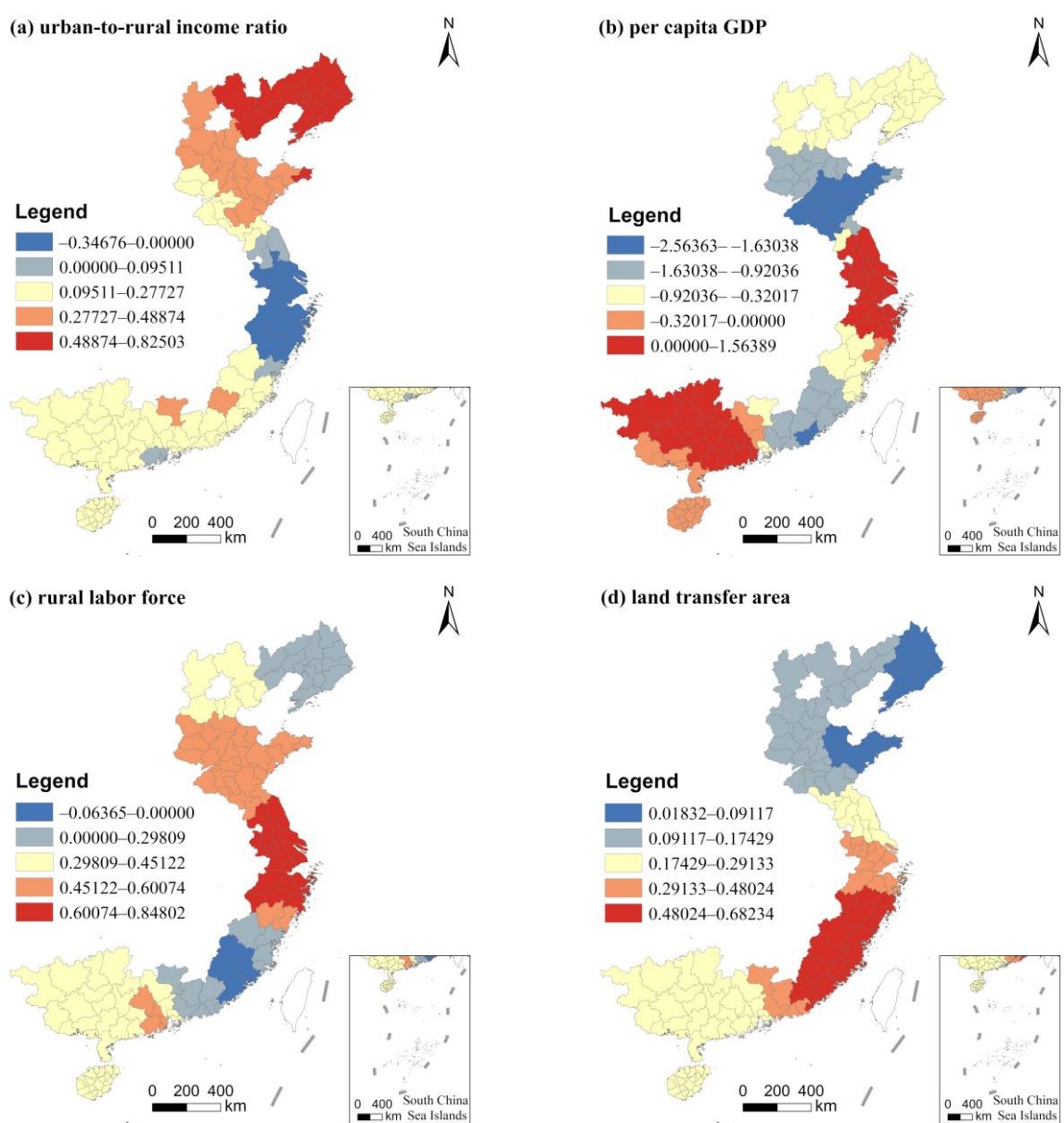

**Figure 7.** Spatial distributions of the regression coefficients between the driving factors and NGP in 130 cities of coastal China-based GWR models. (**a**) urban-to-rural income ratio; (**b**) per capita GDP; (**c**) rural labor force; (**d**) land transfer area.

## 4. Discussion

### 4.1. Analysis of the Causes of NGP and Regional Differentiation in Coastal China

Previous research posits that the low economic efficiency of grain planting [39], the intervention of industrial and commercial capital [11], resource endowment [12–14], and land transfer have driven NGP in China. However, previous studies have mostly focused on macro- or policy factors. This study explores the formation mechanism of NGP from a spatial perspective. The ratio of urban and rural disposable income representing the low economic returns of grain planting, the actual amount of foreign investment in agriculture representing the intervention of industrial and commercial capital, the per capita GDP representing the economic level, and the land transfer system representing policy factors are the main driving forces for the spatial heterogeneity of NGP in coastal China. Although the NGP rate in coastal China is relatively high, there are great differences among provinces. The NGP rate of Liaoning is only 15.6%, while the NGP rate of Hainan is 59.87%. The spatial heterogeneity of the NGP area and NGP rate is more significant at

a municipal scale, which is mainly attributable to differences in the economic, social, policy and resource endowments of each unit. The regression coefficients of the influencing factors based on the GWR model, as shown in Figure 7, revealed the effects of various driving factors of NGP to varying degrees in different cities of coastal China.

In addition, Hainan, Guangdong, Guangxi and Fujian are located in tropical and subtropical zones, with quite favorable climatological conditions for the plantation of agricultural products and fruits, thereby resulting in a more widespread NGP phenomenon. Taking Hainan as an example, its regional climate is still over 20 °C when most of the regions of China are in the middle of a cold winter. Accordingly, Hainan is the most suitable candidate as China's vegetable-production base. Moreover, a stable production-logistics-sales network has been established for some time. Meanwhile, NGP is quite ubiquitous in some hilly areas, which can be regarded as the result of inconvenient traffic flow, low-cost land transfer, and a lack of effective supervision of land use control. Zhu et al. found that convenient transportation links led to the high cost of land conversion, while the use of agricultural machinery and rising energy prices resulted in a greater cost of farming; the combined action of the above two factors caused a decline in the economic benefits of the planting industry [23]. For some villages and towns, both land transfer costs and agricultural labor costs are low because of relatively inconvenient transportation, which may instead increase the planting yield and promote a higher land transfer rate, meanwhile enhancing the probability of NGP on arable land. The conclusion also supports the present research results.

### 4.2. The Potential Impact of NGP on China's Food Security

Coastal China is the most economically developed area of the country. The industrial and commercial capital in this area is constraining agriculture and rural areas, and the proportion of land transfer area to the total cultivated area is much higher than the national average level [73]. Thus, its NGP rate is as high as 33.65%. For example, in Shanghai, the main types of NGP are vegetables and sightseeing agriculture. This is related to Shanghai's special status as the economic center of China, and agriculture is completely subordinate to urban development. The NGP rate in this study area is higher than the mean value of China (27.0%) [39], and significantly exceeded the values of traditional grain-producing areas [59]. However, this is compatible with the level of economic development in coastal China. In fact, China's cultivated land per capita is only 0.09 ha. [74]. In order to meet the demand for rice, wheat and corn, approximately 118 million ha. of cultivated land is required, which accounts for 90.0% of the total cultivated land in China [75]. The NGP in coastal China is much higher than 10.0%. Therefore, coastal China has become the most important area in terms of diversified agricultural products and food imports [76].

Planting diversity may have a positive effect on the ecological environment, especially small-scale afforestation [77–79]. However, the purpose of the farmers' adoption of NGP in China has nothing to do with maintaining the sustainability of the agricultural spatial pattern and structure. The NGP behavior is divided into economic incentives (such as lucrative planting) and policy incentives (such as abandonment and afforestation) in China. The NGP in China is completely different from planting diversification in some developing countries of South Asia and Africa, which is intended more for diversifying natural risks and using land resources efficiently. Some NGP behaviors have harmed the soil fertility of cultivated land, such as the abandonment of farming and afforestation. In addition, digging ponds to farm fish has directly destroyed the fertile layer. Therefore, the government's banning of NGP is intended for long-term food security in China.

### 4.3. How to Effectively Control NGP in Coastal China

According to the causes behind NGP, we should proceed, bearing in mind the following aspects, to effectively control NGP and ensure national food security. (1) The

unmonitored spread of NGP should be controlled and given a scientifically set bottom line. The diversified demands of agricultural products determine the inevitable trend of NGP. China's government should take into account multiple requirements, including future grain demand, natural conditions, ecological protection, and rural revitalization. This kind of assessment should carefully measure the minimum requirements of various types of food and agricultural products from top to bottom, and scientifically set the bottom line of NGP. (2) The external pressure on grain-growing should be reduced and the government should increase farmers' willingness to continue grain-growing. China's government should optimize the regional allocation of agricultural resources, increase investment in agricultural infrastructure, and make high-standard basic farmland. Then, current agricultural subsidy policies should be revised and the traditional subsidy policy according to cultivated area should be abandoned. The agricultural subsidies only provide incentives for grain growers. Meanwhile, the subsidies for NGP should be reduced or canceled, so as to enhance the grain cultivation initiative and remedy the low economic benefits to grain cultivation. (3) The local government should normalize land transfer behaviors, especially the involvement of industrial and commercial capital, preventing free-riding or touching-ball behaviors. Agricultural and natural resource management departments should implement the full supervision of cultivated land that is transferred and should severely punish unauthorized changes of cultivated land use or the destruction of cultivated land, in accordance with the law, and thoroughly curb NGP caused by land transfer.

In addition, based on various statistical data, this study attempted to determine the explanatory contributions of various factors on NGP behaviors in coastal China and establish spatial heterogeneity with the related model. However, the change of NGP on cultivated land is as a result of the combined action of various factors, which should be described in terms of a complex nonlinear relationship rather than a simple positive or negative correlation. Only relying on a particular model may cause deviation in the knowledge of NGP, and the changing processes, the environmental effects, and the mechanism of NGP should be further explored in future studies [73,76,80,81].

## 5. Conclusions

Food security is seriously threatened by NGP in coastal China. This study takes 13 provinces and 130 cities in coastal China as research objects and explores the spatial patterns and distinguishing characteristics of NGP, as recorded in 2018. The results are as follows. (1) The NGP area in coastal China in 2018 reached up to $15.82 \times 10^6$ hm$^2$, occupying 33.65% of the total cultivation area. Nearly one-third of the cultivated land has already been occupied by NGP. (2) The NGP area and rate in coastal China showed different spatial agglomeration characteristics. Overall, the NGP rate dropped gradually from south to north, but had two peaks at Shandong and Guangxi and then weakened toward the surrounding area. (3) The low economic benefit of crop production (the ratio of per capita GDP to urban and rural disposable income) is the most significant driving force of NGP on arable land in coastal China. The social factor (the amount of rural labor force) and the policy factor (land transfer area) are the main driving factors that accounted for the differentiation in the spatial distribution pattern of NGP. However, the driving factors had an obvious spatial heterogeneity.

**Author Contributions:** Conceptualization, F.C.; data curation, X.G. and J.L.; funding acquisition, F.C.; investigation, J.L. and Y.C.; methodology, J.L. and Y.C.; project administration, F.C.; software, X.G. and Y.S.; supervision, F.C.; visualization, G.-J.L.; writing—original draft, Y.S. and Y.C.; writing—review and editing, G.-J.L. and F.C. All authors have read and agreed to the published version of the manuscript.

**Funding:** This work was supported by the Key Projects in the National Science and Technology Pillar Program during the Twelfth Five-year Plan Period (2015BAD06B02). In addition, the authors

would like to thank the Institute of Land Surveying and Planning of Jiangsu for their support during the research. There is no conflict of interest in this manuscript.

**Institutional Review Board Statement:** Not applicable.

**Informed Consent Statement:** Not applicable.

**Data Availability Statement:** Not applicable.

**Conflicts of Interest:** The authors declare no conflict of interest.

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
