# Peer review of "Spatial Differentiation of Non-Grain Production on Cultivated Land and Its Driving Factors in Coastal China"

_sustainability, doi:10.3390/su132313064_

Round 1
Reviewer 1 Report
The research idea is interesting, the layout is correct and the conclusions come from the analysis. However, in my opinion, the discussion should be expanded to include aspects related to the sustainable development. Keeping a sustainable both spatial pattern and structure of crops is important for economic, social and environmental reasons. In this context, it should be considered whether a certain share of non-grain production on cultivated land is advantageous in some respects, despite its negative impact on food production. For example, even small-scale afforestation may have a positive effect on combating climate change (although due to the scale of the phenomenon, this impact may not be measurable). Of course, spatial pattern of NGP and its causes are important for the current precise cultivated land protection and the national food security. However, the importance of spatial pattern for sustainable development is at least as important and this aspect cannot be omitted in the analysis and recommendations.

Author Response
Thank you very much for your great efforts on our manuscript. We have revised the manuscript according to your suggestions and comments (marked in red color). The scope of discussion was expanded to include the possible social, economic, ecological and sustainable impacts of planting diversity. See lines 487-497 for specific modifications. The details of our revisions are as follows.
Planting diversity may have a positive effect on the ecological environment, especially small-scale afforestation [75-77]. However, the purpose of farmer’s NGP in China has nothing to do with maintaining the sustainability of the agricultural spatial pattern and structure. The NGP behavior is divided into economic incentives (such as cash planting) and policy incentives (such as abandonment and afforestation) in China. The NGP in China is completely different from planting diversification in South Asian countries, which is more for diversifying natural risks and using land resources efficiently. Some NGP behaviors have harmed the soil fertility of cultivated land, such as abandoned farming and afforestation. In addition, digging ponds to raise fish directly destroyed the farming layer. Therefore, the government's banning of NGP is for long-term food security in China.
Reviewer 2 Report
As China has more people and less land, the cultivated land protection occupies an extremely important position in China's food security strategy. Due to the impact of comparative benefits brought about by the differences in urban and rural economic development, some non-graining phenomena that are contrary to traditional planting concepts have appeared in China. If this phenomenon is not stopped, it may affect China's food security to some extent. Therefore, this research has certain theoretical and practical value. Overall, the article is rich in content, clear in thinking, rigorous in logic, and has a certain degree of innovation and reference significance. However, it still needs to be fully revised before publication. The detailed comments are as follows,
(1) The literature review in the second paragraph of the introduction should re-organize the logic and present it in sections to improve readability. The last paragraph supplements the typicality of study area selection.
(2) In the material method section, L108 mentioned "The definition and calculation methods of NGP have always been very convincing in China". As far as I know, there is no unified conclusion on non-grain calculations in China. Actually, detection by long-time sequence remote sensing seems to be the most widely used. So, why doesn't the author use remote sensing methods or statistical data methods?
(3) The data application of some NGPs in 2.1.1 is not clear enough, for example, how is the multiple index obtained?
(4) L120, "The advantages of the definition of NGP in this study are has two advantages", there are problems with the expression, and similar sentences must be checked.
(5) The article lacks an overview of the region, which prevents non-local readers from having an overall understanding of the study area.
(6) The presentation of the result analysis is a bit verbose. Especially the listing of different cities in section 3.1 is meaningless. It is advisable to simply analyze the content of the result, such as the general reason for the distribution in certain areas, rather than always saying where the distribution is. Besides, non-local readers may not know where the city you are talking about is. As a digression, it is recommended to add the name of the area when describing the regional overview.
(7) L300, the Moran index value seems to be a bit too low, does it mean that the non-graining of the region may not have too strong spatial correlation? If the correlation is not strong, subsequent data analysis is actually of no value.
(8) L372-373, "If the value of AICc in GWR-based fitting results exceeds the value in OLS-based regression results, GWR exceeds OLR regression in regression fitting performance and applicability." the expression of this sentence is contrary to the results presented in Table 3.
(9) Is the non-grain conversion rate of 33.65% credible? It is recommended to supplement the comparison of related existing research results in the discussion section.
(10) The discussion part is still relatively shallow, and there is still room for improvement. On the one hand, the rate of non-graining affects the safety of rations, but it may bring about the improvement of the ecological environment. On the other hand, in the context of China's grain storage strategy, as a means of agricultural structural adjustment, whether the rate of non-graining will really affect food security is also unknown. It is recommended that the author go further and supplement the discussion content.
(11) In the conclusion part, there is no need to talk about the background, and directly cut into the key and innovative results, highlighting the content that is different from other studies. In addition, the policy recommendations in 5.2 should be put into the discussion.
(12) L517, many phrases are difficult to understand. It is recommended that native speakers modify the language of this article.
Author Response
As China has more people and less land, the cultivated land protection occupies an extremely important position in China's food security strategy. Due to the impact of comparative benefits brought about by the differences in urban and rural economic development, some non-graining phenomena that are contrary to traditional planting concepts have appeared in China. If this phenomenon is not stopped, it may affect China's food security to some extent. Therefore, this research has certain theoretical and practical value. Overall, the article is rich in content, clear in thinking, rigorous in logic, and has a certain degree of innovation and reference significance. However, it still needs to be fully revised before publication. The detailed comments are as follows:
(1) The literature review in the second paragraph of the introduction should re-organize the logic and present it in sections to improve readability. The last paragraph supplements the typicality of study area selection.
Thank you very much for your suggestions. (1) We reorganized the literature review and presented it in the form of sections. See lines 51-99 for details. As follows:
NGP has been a common phenomenon of land use conversion on a global scale. NGP is different from non-agriculturalization, does not cause loss of cultivated land [14], and is different from farmland marginalization [15]. Generally, NGP refers to the phenomenon that the cultivated land for grain cultivation is occupied by cash crops, forests, fruits or the breeding industry. NGP is mainly attributed to low economic benefit of planting industry, and is also affected by regional resource endowments, economic development, social factors and agricultural production habits. European and American countries rarely intervene in land use conversion within the agricultural sector, and mainly rely on the market to adjust the planting structure. The United States, based on the international and domestic food supply and demand conditions, formulated the lower limit of the planting scale of major crops every year, which not only guaranteed food security, but also fully protected the farmland ecology and high-efficiency utilization of farmland resources [16]. EU countries and Japan adjusted their fallow scale according to market supply and demand conditions to cope with fluctuations in international food prices [17-20]. The agricultural planting structure of some developing countries such as Brazil is not only affected by their own needs, but also by the investment motives of multinational companies [21]. The Chinese government discourages NGP, because it needs to use 8% of the world’s cultivated land to feed 18.4% of the world’s population.
Food security is the primary consideration of the Chinese government. The Chinese government has long discouraged NGP and issued an order prohibiting NGP on November 17, 2020. However, it is very difficult to enforce the policy ban of NGP. Firstly, both local governments and farmers lack motivation because of low economic benefits in crop planation. Some researchers have validated the drive function of low economic benefits of crop plantation on NGP and analyzed the negative effect of NGP on national food security [22-25]. Secondly, land transfer and profit-seeking of industrial and commercial capital promoted the conversion to NGP. Seto et al. found that foreign investment pulling, unscientific planning, and improper management have led serious loss of cultivated land and grain production capability in Pearl River Delta [26]. Liao et al. analyzed the effects of profit-seeking behaviors of contractors on NGP in land transfer process and proposed the related measures and suggestions of preventing excess NGP from the perspective of restricting improper land transfer [27]. Thirdly, unlike non-agriculturization, NGP did not directly lead to the loss of cultivated lands, with less hazard. NGP can be mostly regarded as internal structural adjustment of agriculture and cannot be completely prohibited because of the involvement of farmers’ interests. In addition, some specific factors including cultivation scale [28-30], industrial structure [31], family characteristics [32-34] and benefit farming policies [35] also affected NGP in China.
Remote sensing and UAV technology have been provided an effective means for monitoring NGP [36]. Some studies have focused on the spatial characteristics of cultivated land, non-agriculturalization, and farmland marginalization [37, 38]. However, its workload and cost are huge. Remote sensing data with a resolution greater than 1 m is also difficult to meet some special crop monitoring, such as garlic and wheat cannot be distinguished at all in winter. Currently, it was almost impossible to use remote sensing technology to accurately monitor NGP on a national scale as large as China. Authoritative statistical data is indispensable for studying l the spatial characteristics of NGP at a national scale. Moreover, cultivated land is widely distributed, with large differences in natural, social and economic conditions in China. The behavior and causes of NGP in different regions are not the same. At a large scale, clarify the spatial pattern of NGP and its causes for the current precise cultivated land protection and the national food security is particularly important.
(2) The typicality of the selected study area is explained in addition, as detailed in Lines 106 to 109.As follows:
Therefore, 11 provinces and 130 cities in coastal China were selected as the research objects in this study, because they are most severely affected by the low economic efficiency of grain planting, the intervention of industrial and commercial capital, and urbanization.
(2) In the material method section, L108 mentioned "The definition and calculation methods of NGP have always been very convincing in China". As far as I know, there is no unified conclusion on non-grain calculations in China. Actually, detection by long-time sequence remote sensing seems to be the most widely used. So, why doesn't the author use remote sensing methods or statistical data methods?
Thank you very much for the suggestions of reviewers. We have corrected the presentation error (See lines 117), and made corresponding explanations on why remote sensing monitoring was not used for our study. See lines 87-95 for details. As follows:
Lines 117: The definition and calculation methods of NGP have always been very vague in China.
Lines 87-95: Remote sensing and UAV technology have been provided an effective means for monitoring NGP [36]. Some studies have focused on the spatial characteristics of cultivated land, non-agriculturalization, and farmland marginalization [37, 38]. However, its workload and cost are huge. Remote sensing data with a resolution greater than 1 m is also difficult to meet some special crop monitoring, such as garlic and wheat cannot be distinguished at all in winter. Currently, it was almost impossible to use remote sensing technology to accurately monitor NGP on a national scale as large as China. Authoritative statistical data is indispensable for studying l the spatial characteristics of NGP at a national scale.
(3) The data application of some NGPs in 2.1.1 is not clear enough, for example, how is the multiple index obtained?
Thank you very much for your suggestions. (1) We have revised the description of the data source. See lines 129-131 for details. As follows:
Lines 129-131: The cultivated area, the grain sowing area, and the multiple crop index of municipal unit are directly derived from the 2019 statistical yearbooks of each municipal unit.
(4) L120, "The advantages of the definition of NGP in this study are has two advantages", there are problems with the expression, and similar sentences must be checked.
Thank you very much for the suggestions of reviewers. We have corrected the mistakes. See lines 131-137 for details. The following:
The advantage of the definition of NGP in this study is that the aforementioned three indicators have accurate and authoritative statistical data in China, and at the same time, it avoids the existence of arable land and facility agricultural land in vegetable production, and the existence of arable land and garden land in fruit production. Therefore, we can only rely on authoritative statistical data to monitor NGP in China. In addition, considering the multiple cropping index also makes the rate of NGP closer to the actual situation.
(5) The article lacks an overview of the region, which prevents non-local readers from having an overall understanding of the study area.
Thank you very much for the suggestions of reviewers. We have made corresponding modifications. (1) The text description of the study area is added. See lines 155-161 for details. (2) In addition, the location map of the study area was drawn and the name of the study area was added. See Figure 2 in the attachment for details. As follows:
Lines 155-161:
2.2.1. The study area
Coastal China has 11 provincial units, including Liaoning, Hebei, Tianjin, Shandong, Jiangsu, Shanghai, Zhejiang, Fujian, Guangdong, Guangxi, and Hainan, with a total area of 950,000 km2 (Fig. 2). The coastal area straddles the tropical, subtropical and temperate zones, and belongs to a typical maritime monsoon climate. The annual precipitation is 750-1,700 mm, decreasing from south to north. In 2020, the coastal area will have a population of approximately 643 million, accounting for 44.5% of the total population in China, with an urbanization rate of over 60.0% in 130 municipal units, and a per capita GDP close to $20,000, almost including all of China's most economically developed provincial units. The per capita cultivated land is small, only 0.08 ha in coastal China. Affected by the low economic efficiency of the planting industry, the intervention of industrial and commercial capital, and urbanization, the situation of grain planting in coastal China is not good, and the phenomenon of NGP has become very common.
(6) The presentation of the result analysis is a bit verbose. Especially the listing of different cities in section 3.1 is meaningless. It is advisable to simply analyze the content of the result, such as the general reason for the distribution in certain areas, rather than always saying where the distribution is. Besides, non-local readers may not know where the city you are talking about is. As a digression, it is recommended to add the name of the area when describing the regional overview.
Thank you very much for the suggestions of reviewers. We have made corresponding modifications. The analysis results are briefly summarized and analyzed. See lines 303-313 for details. As follows:
In terms of NGP area, 11 cities can be classified as high-value cities, which were dispersed in Guangxi, Shandong and Liaoning. 39 cities can be classified as low-value cities, which were relatively concentrated in the eastern Liaoning, the coastal areas of Zhejiang and Fujian, the Pearl River Delta, and Hainan (Fig.4a). Secondly, NGP rate also dropped from south to north with greater spatial heterogeneity at municipal scale. NGP rates of 31 municipal cites all exceeded 58%. These 31 municipal units, with NGP rate of over 58%, exhibited concentrated distribution among the East China Sea (Fig.4b). These units were mostly located in hilly areas, with small cultivation plots and per capita cultivation area, which were unfavorable for mechanized operation. In addition, favorable climatic conditions were suitable for the planation of tropical and sub-tropical fruits.
(7) L300, the Moran index value seems to be a bit too low, does it mean that the non-graining of the region may not have too strong spatial correlation? If the correlation is not strong, subsequent data analysis is actually of no value.
Thanks a lot for the suggestions of reviewers. Although the Moran index is low, it has passed the significance test (P < 0.001), indicating that there is a strong spatial correlation between the non-grain conversion of cultivated land in the research region. See lines 324-325 for details. The following:
The Global Moran indexes () of NGP area and rate of 130 cities were 0.229 and 0.394, respectively, which exceeded 0 and passed the significance level test at P<0.001.
(8) L372-373, "If the value of AICc in GWR-based fitting results exceeds the value in OLS-based regression results, GWR exceeds OLR regression in regression fitting performance and applicability." the expression of this sentence is contrary to the results presented in Table 3.
Thank you very much for the suggestions of reviewers. We have corrected the mistakes. Please refer to Line 396-398 for details. The following:
If the value of AICc in GWR-based fitting results is lower than the value in OLS-based regression results, GWR is better than OLR regression in regression fitting performance and applicability.
(9) Is the non-grain conversion rate of 33.65% credible? It is recommended to supplement the comparison of related existing research results in the discussion section.
Thank you very much for the suggestions of reviewers. We have made corresponding modifications. References are cited and comparisons of existing research results are added to illustrate the credibility of the 33.65% non-grain conversion rate in coastal areas. See lines 479-481 for details. As follows:
The NGP rate in this study area is higher than the mean value of China (27.0%) [39], and significantly exceeded the values of traditional grain producing areas [59]. However, this is compatible with the level of economic development in coastal China.
(10) The discussion part is still relatively shallow, and there is still room for improvement. On the one hand, the rate of non-graining affects the safety of rations, but it may bring about the improvement of the ecological environment. On the other hand, in the context of China's grain storage strategy, as a means of agricultural structural adjustment, whether the rate of non-graining will really affect food security is also unknown. It is recommended that the author go further and supplement the discussion content.
Thank you very much for the suggestions of reviewers. We have supplemented and improved the contents of the discussion section according to the suggestions of reviewers. See lines 440-454 for details. As follows:
Previous research believes that the low economic efficiency of grain planting [39], the intervention of industrial and commercial capital [11], resource endowment [12-14], and land transfer have driven NGP in China. However, previous studies have mostly focused on macro- or policy factors. This study explores the formation mechanism of NGP from a spatial perspective. The ratio of urban and rural disposable income representing the low economic returns of grain planting, the actual amount of foreign investment in agriculture representing intervention of industrial and commercial capital, the per capita GDP representing the economic level, and the land transfer representing policy factors are the main driving forces for spatial heterogeneity of NGP in coastal China. Although the NGP rate in coastal China is relatively high, there are great differences among provinces. The NGP rate of Liaoning is only 15.6%, while the NGP rate of Hainan is 59.87%. Spatial heterogeneity of NGP area and NGP rate is more significant at a municipal scale, which is mainly attributed to the differences in the economic, social, policy and resource endowments of each unit. The regression coefficients of the influencing factors based on GWR model, as shown in Fig. 6, revealed the effects of various driving factors of NGP to varying degrees in different cities of coastal China.
(11) In the conclusion part, there is no need to talk about the background, and directly cut into the key and innovative results, highlighting the content that is different from other studies. In addition, the policy recommendations in 5.2 should be put into the discussion.
Thank you very much for the suggestions of reviewers. We have made corresponding modifications. Remove context, simplify conclusions, and incorporate policy recommendations into the discussion. See lines 472-520 for details. As follows:
4.2. The potential impact of NGP on China's food security and other
Coastal China is the most economically developed areas. The industrial and commercial capital in this area is more squeezing agriculture and rural areas, and the proportion of land transfer area to the total cultivated area is much higher than the national average level. So, its NGP rate is as high as 33.65%. For example, in Shanghai, the main types of NGP are vegetables and sightseeing agriculture. This is related to Shanghai's special status as the economic center of China, and agriculture is completely subordinate to urban development. The NGP rate in this study area is higher than the mean value of China (27.0%) [39], and significantly exceeded the values of traditional grain producing areas [59]. However, this is compatible with the level of economic development in coastal China. In fact, China's per capita cultivated land is only 0.09 ha [73]. In order to meet the needs of rice, wheat and corn, approximately 118 million ha of cultivated land must be required, which accounts for 90.0% of the total cultivated land in China [74]. The NGP in coastal China is much higher than 10.0%. Therefore, coastal China has become the most important area for diversified agricultural products and food imports.
Planting diversity may have a positive effect on the ecological environment, especially small-scale afforestation [75-77]. However, the purpose of farmer’s NGP in China has nothing to do with maintaining the sustainability of the agricultural spatial pattern and structure. The NGP behavior is divided into economic incentives (such as cash planting) and policy incentives (such as abandonment and afforestation) in China. The NGP in China is completely different from planting diversification in South Asian countries, which is more for diversifying natural risks and using land resources efficiently. Some NGP behaviors have harmed the soil fertility of cultivated land, such as abandoned farming and afforestation. In addition, digging ponds to raise fish directly destroyed the farming layer. Therefore, the government's banning of NGP is for long-term food security in China.
4.3. How to effectively control NGP in coastal China
According to the causes of NGP, we should proceed from the following aspects to effectively control NGP and ensure national food security. (1) Control unordered spread and scientifically set bottom line of NGP. Diversified demands of agricultural products determine the inevitable trend of NGP. The China’s government should take into account multiple requirements including future grain demand, natural condition, ecological protection, and rural revitalization. This kind of assessment should carefully measure minimum requirements of various types of food and agricultural products from top to bottom, and scientifically set the bottom line of NGP; (2) Reduce the external pressure of grain-growing and increase farmers’ willingness of grain-growing. The China’s government should optimize regional allocation of agricultural resources, increase investment in agricultural infrastructure, and build high-standard basic farmland. Then, current agricultural subsidy policies should be revised and abandon traditional subsidy way in accordance with cultivated area. The agricultural subsidies only provide to grain growers. Meanwhile, the subsidies for NGP should be reduced or cancelled so as to enhance grain cultivation initiative and remedy low-economic benefits in grain cultivation; (3) Normalize land transfer behavior and compact the responsibility of land users. The local government should normalize land transfer behaviors, especially the involvement of industrial and commercial capital, prevention free-riding or touching-ball behaviors. Agricultural and natural resource management departments should implement full supervision of cultivated land that is transferred, and severely punish the unauthorized changes of cultivated land or the destruction of cultivated land in accordance with the law, and thoroughly curb NGP caused by land transfer.
(12) L517, many phrases are difficult to understand. It is recommended that native speakers modify the language of this article.
Thank you very much for the suggestions of reviewers. We have corrected some mistakes. See lines 513-516 for details. The following:
Normalize land transfer behavior and compact the responsibility of land users. The local government should normalize land transfer behaviors, especially the involvement of industrial and commercial capital, prevention free-riding or touching-ball behaviors.

Round 2
Reviewer 2 Report
The article has been greatly improved, and it is recommended that the author add some literature (on the environmental effects of NGP) to increase the possibility of future research in the discussion section. Some for reference:
Koch, J., Schaldach, R., Göpel, J., 2019. Can agricultural intensification help to conserve biodiversity? A scenario study for the African continent. Journal of Environmental Management, 247, 29–37. DOI: 10.1016/j. jenvman.2019.06.015
Liang, X., Jin, X., Sun, R., Bo, H., Liu, J., Zhou, Y. 2021. A typical phenomenon of cultivated land use in China's economically developed areas: Anti-intensification in Jiangsu Province . Land Use Policy, 102, 105223. DOI: 10.1016/j.landusepol.2020.105223
Long, H., Qu, Y. 2018. Land use transitions and land management: A mutual feedback perspective. Land Use Policy, 74, 111–120. DOI: 10.1016/j.landusepol.2017.03.021
Zabel, F., Delzeit, R., Schneider, JM, Seppelt, R., Mauser, W., Václavík, T., 2019. Global impacts of future cropland expansion and intensification on agricultural markets and biodiversity. Nature Communications, 10( 1), 2844. DOI: 10.1038/s41467-019-10775-z
